# Structural analysis of viral ExoN domains reveals polyphyletic hijacking events

Adrián Cruz-González[1], Israel Muñoz-Velasco[1], Wolfgang Cottom-Salas[1,2], Arturo Becerra[1], José A. Campillo-Balderas[1], Ricardo Hernández-Morales[1], Alberto Vázquez-Salazar[3], Rodrigo Jácome[1]*, Antonio Lazcano[1,4]*

**1** Facultad de Ciencias, Universidad Nacional Autónoma de México, México City, México, **2** Escuela Nacional Preparatoria, Plantel 8 Miguel E. Schulz, Universidad Nacional Autónoma de México, México City, México, **3** Department of Chemical and Biomolecular Engineering, University of California, Los Angeles, California, United States of America, **4** El Colegio Nacional, México City, México

* alar@ciencias.unam.mx (AL); rodrigo.jacome@ciencias.unam.mx (RJ)

**Data Availability Statement:** All relevant data are within the manuscript and its Supporting information files.

## Abstract

Nidoviruses and arenaviruses are the only known RNA viruses encoding a 3'-5' exonuclease domain (ExoN). The proofreading activity of the ExoN domain has played a key role in the growth of nidoviral genomes, while in arenaviruses this domain partakes in the suppression of the host innate immune signaling. Sequence and structural homology analyses suggest that these proteins have been hijacked from cellular hosts many times. Analysis of the available nidoviral ExoN sequences reveals a high conservation level comparable to that of the viral RNA-dependent RNA polymerases (RdRp), which are the most conserved viral proteins. Two highly preserved zinc fingers are present in all nidoviral exonucleases, while in the arenaviral protein only one zinc finger can be identified. This is in sharp contrast with the reported lack of zinc fingers in cellular ExoNs, and opens the possibility of therapeutic strategies in the struggle against COVID-19.

## Introduction

As of today, the coronavirus SARS-CoV-2 pandemic has affected more than 100 million people worldwide, causing millions of deaths, as well as a severe sanitary, social, and economic crisis [1]. The *Coronaviridae* family is part of the *Nidovirales* order, which includes enveloped, non-segmented and single positive-stranded RNA (+ssRNA) viruses that infect a wide variety of animal hosts, including humans [2–7].

As in most nidoviruses, the coronaviral genome has two large ORFs (ORF1a and ORF1b), followed first by a set of four structural protein genes: spike (S), membrane (M), envelope (E), and nucleocapsid (N), and then by a varying number of ORFs encoding the so-called accessory proteins [8, 9]. Translation of ORF1a and ORF1b produces two polyproteins, polyprotein 1a (pp1a) and polyprotein 1ab (pp1ab), which result in sixteen non-structural proteins (nsp 1–16) involved in the delivery of viral progeny [10]. Pp1a harbors nsp1–11, whereas pp1ab contains nsp1–16, which results from a programmed –1 ribosomal frameshift at a short overlap of ORF1a with ORF1b [11]. Nidoviruses exhibit considerable diversity in terms of the

**Funding:** All the authors we would like to acknowledge financial support of DGAPA-UNAM (PAPIIT-IN214421 and PAPIME-PE204921).

**Competing interests:** The authors have declared that no competing interests exist.

number and size of proteins encoded in their genomes. Nevertheless, five core domains are conserved among them, including the main protease (Mpro), the RNA-dependent RNA polymerase (RdRp), the RdRp-associated nucleotidyltransferase (NiRAN), the superfamily 1 helicase domain (HEL1), and a zinc-binding domain (ZBD), which is always associated with HEL1 [12].

The largest known RNA viral non-segmented genomes are found in nidoviruses. The upper limit is held by the Planarian secretory-cell nidovirus (PSCNV), with a 41.1 kb genome [13, 14]. The presence of these unusually long linear RNA-genomes is explained in part by the proofreading activity of their 3'-5' exonuclease (ExoN) domain [12, 15], which hydrolyzes phosphodiester bonds to cleave nucleotides of a polynucleotide chain (both ssRNA and dsRNA) when they are misincorporated at the 3' end during the replication process [16, 17]. At present, only eight of the fourteen families recognized by the International Committee on Taxonomy of Viruses in the *Nidovirales* order (*Coronaviridae, Tobaniviridae, Roniviridae, Medioniviridae, Euroniviridae, Mesoniviridae, Abyssoviridae,* and *Mononiviridae*), all of which possess genomes of 20 kb or larger (Table 1), are known to be endowed with the proofreading ExoN domain.

The coronaviral nsp14 protein is composed of two different functional domains, the N-terminus which corresponds to the ExoN, and the C-terminal domain which is an N7-methyltransferase (N7-MTase) that caps the RNA avoiding its degradation [18]. The importance of the ExoN domain has been corroborated experimentally. In ExoN-knockout mutants of the betacoronaviruses Mouse hepatitis virus (MHV) and SARS-CoV, replication fidelity is strongly diminished, conferring them with a "mutator phenotype" viable in cell culture [19, 20]. Moreover, inactivation of the ExoN of HCoV-229E (alphacoronavirus), MERS-CoV, and SARS-CoV-2 (betacoronaviruses) severely affects the replication process, and results in failure to recover infectious viral progeny [16, 21].

Based on sequence comparisons, Snijder et al. (2003) demonstrated the evolutionary relationship between the coronaviral ExoN domain and the cellular DNA-proofreading enzymes of the DEDD (DnaQ-like) family of exonucleases [22]. Several features are shared among the coronaviral ExoN and the DnaQ-like family of exonucleases, including the well-conserved

**Table 1. Viral families belonging to *Nidovirales* order.**

| Suborder | Family | Genome size (kb) | Hosts |
|---|---|---|---|
| *Amidovirineae* | *Arteriviridae* | 12.7–15.7 | Horses, pigs, possums, shrews, primates, rodents |
| | *Cremegaviridae* | ? | unknown? |
| | *Gresnaviridae* | 18.4 | Snake |
| | *Olifoviridae* | 15.3 | Snake |
| *Nanidovirineae* | *Nanghoshaviridae* | 13.1 | Fishes |
| | *Nanhypoviridae* | 18.2 | Fishes |
| *Mesnidovirineae* | *Medioniviridae* | ~ 20.2–25 | Gastropods |
| | *Mesoniviridae* | ~ 20 | Arthropods |
| *Cornidovirineae* | Coronaviridae | ~ 27–32 | Birds, cattles, dogs, cats, pigs, rodents, whales, humans |
| *Tornidovirineae* | *Tobaniviridae* | ~ 20–33 | Mammals, fishes, snakes |
| *Ronidovirineae* | *Roniviridae* | ~ 26 | Fishes, shrimps |
| | *Euroniviridae* | ~ 24.5 | Crustaceans |
| *Abnidovirineae* | *Abyssoviridae* | 35.9 | Gastropods |
| *Monidovirineae* | *Mononiviridae* | 41.1 | Helminthes |

Blue, viruses lacking the ExoN domain; red, viruses endowed with the ExoN domain.

DEDD motif, the 3'-5' exonucleolytic degradation of DNA and/or RNA, and the β1-β2-β3-αA-β4-αB-β5-αC conserved catalytic core topology [23]. Homologs of the DnaQ-like family of exonucleases have also been identified in other viruses, including the single-stranded negative RNA arenaviruses, as well as in the double-stranded DNA φ-29, T4 and T7 phages [24–28].

A noteworthy feature of the coronaviral nsp14 is that its two protein domains are endowed with zinc fingers (ZFs), two in the case of ExoN and one in the case of N7-MTase [18, 29]. ZFs can be described as a group of stable scaffolds whose structure is maintained by the zinc ion. They vary in sequence and structure, which reflect the $Zn^{2+}$ ion coordination with cysteine and/or histidine residues and the way in which the ZF interacts with other molecules [30–32]. Typically, ZFs act as interaction modules and bind to several molecules, including nucleic acids, proteins, lipids, and small compounds [32, 33]. The distinctive ZFs of the SARS-CoV ExoN domain appear to play a key role in the structural stability of the enzyme [18]. According to the Andreini et al. (2011) zinc sites classification, the SARS-CoV ExoN ZF1 and ZF2 are a shuffled zinc ribbon and C2H2, respectively. Although ZFs are found in numerous eukaryotic proteins as well as in the bacterial Ros\MucR protein family [30], they have not been identified in known cellular and dsDNA viral exonucleases.

To the best of our understanding, this is the first evolutionary analysis of the RNA viral exonucleases in which their cellular counterparts have been included demonstrating the polyphyletic viral hijacking of the cellular ExoN gene. Due to the mutation rate disparity between DNA and RNA-based biological entities, we have built tertiary structure-based phylogenies, in which several viral ExoN domain hijacking events can be recognized. Phylogenetic analysis of the available nidoviral ExoN sequences reveals a level of sequence conservation similar to that of the viral RNA-dependent RNA polymerases (RdRp). This and its cornerstone relevance in the viral cycle suggest that the SARS-CoV-2 ExoN should be considered as a therapeutic target in the struggle against the Covid-19 pandemic. Since the cellular ExoNs lack ZFs, our findings suggest that conserved ZFs of the nidoviral ExoN domain might be seen as therapeutic targets in the control of coronaviral infections and for the understanding of the early evolution of this viral order.

## Materials and methods

### Structural comparisons and structure-based tree construction

A search for structural homologs of the SARS coronavirus ExoN domain (nsp14-nsp10 complex, chain B, residues 1–287; PDB ID: 5C8U) was made in the PDB database using the DALI server [34]. Thirty-three non-redundant crystallographic structures with a Z score >4 were collected from the PDB database [35]. To construct the structure-based evolutionary tree, we performed pairwise comparisons between the selected structures with the PDBe Fold online server [36]. From each pairwise comparisons, we collected the root mean-square deviation (RMSD) and the number of superimposed residues to calculate the Structural Alignment Score (SAS) [37], which is defined as:

$$(100 \; X \; RMSD) \; \div \; number \; of \; superimposed \; residues$$

*where RMSD is*

$$\sqrt{\frac{1}{n}\sum_{i=1}^{n} d_i^2}$$

and constructed a geometrical distance matrix using the aforementioned values. Finally, the

program Fitch, included in the PHYLIP package [38], was used to compute the tree with the SAS distance matrix.

### Retrieval of ExoN and RdRp sequences

*Nidovirales* polyprotein 1ab sequences were manually obtained from the NCBI-RefSeq database. For each sequence, a local alignment algorithm was run (Smith-Waterman with default parameters), using the SARS-CoV-2's (YP_009725309.1) ExoN domain and nsp12 (RdRp) sequences as queries. ExoN and RdRp homologous sequences were also retrieved from viral and cellular NCBI-RefSeq database (December 2020) using SARS-CoV-2 NSP14 (YP_009725309.1) and NSP12 (PDB-6M71) as queries, respectively. For this purpose, a Smith-Waterman local alignment was carried out (default parameters).

### Multiple sequence alignment and phylogeny estimation of ExoN and RdRp

A multiple sequence alignment was built for each of the two proteins with the PROMALS3D server [39] uploading the SARS-CoV ExoN domain (PDB 5C8U, [18]) and the SARS-CoV-2 nsp12 (PDB 6M71) tertiary structures as references. For the ExoN sequences, the best model under Akaike criterion was LG+F+R5 calculated with ModelFinder [40], and a phylogeny was inferred with maximum likelihood and 100 non-parametric bootstraps implemented in IQ-TREE2 [41]. Branches with bootstrap values <50% were collapsed using TreeCollapserCL4 [42]. The RdRp alignment was treated with the trimAL program [43] using the automated1 heuristic method. The best model under Akaike criterion was LG+F+R6 calculated with ModelFinder [40], and a phylogeny was inferred with maximum likelihood and 100 non-parametric bootstraps implemented in IQ-TREE2 [41]. Branches with bootstrap values <50% were collapsed using TreeCollapserCL4 [42]. The phylogenies were edited and visualized with Figtree (http://tree.bio.ed.ac.uk/software/figtree/) [44].

## Results

### ExoN tertiary structure-based phylogenetic tree

As expected, the multiple alignment of viral and cellular ExoN's sequences led to a non-conclusive tree that reflected the high degree of divergence within this diverse family that includes proteins encoded by RNA viruses, DNA phages, and cells. Since protein tertiary structure is more conserved than the amino acid sequence [45, 46], we built a phylogenetic tree based on the spatial superpositions of the available tertiary structures of DnaQ-like family exonucleases. Our results confirm the monophyletic origin of all these exonucleases. As shown in Fig 1, there is a non-random distribution of the ExoN domains, with the DEDDh and DEDDy exonucleases located on clearly defined different clades. Exonucleases have a wide array of functions involving RNA and DNA repair, proofreading, immune activity, etc., and the fact that these different functions are located in the same clades highlights their functional versatility and lack of absolute substrate specificity. The SARS-CoV ExoN is found within a branch encompassing eukaryotic and prokaryotic DEDDh ExoNs with multiple functions, including proofreading, cytoplasmic nucleic acid degradation, and RNA maturation, processing and binding activities. On the other hand, the arenaviral ExoNs are located in a different branch of DEDDh ExoNs, and its sister branch includes the MAEL enzymes, which are an atypical group of nucleases lacking the characteristic DEDDh/y catalytic pentad (see below). Finally, the dsDNA phages ExoNs are grouped together with other cellular DEDDy ExoNs, all of which partake in DNA proofreading. This tree strongly suggests that these monophyletic viral

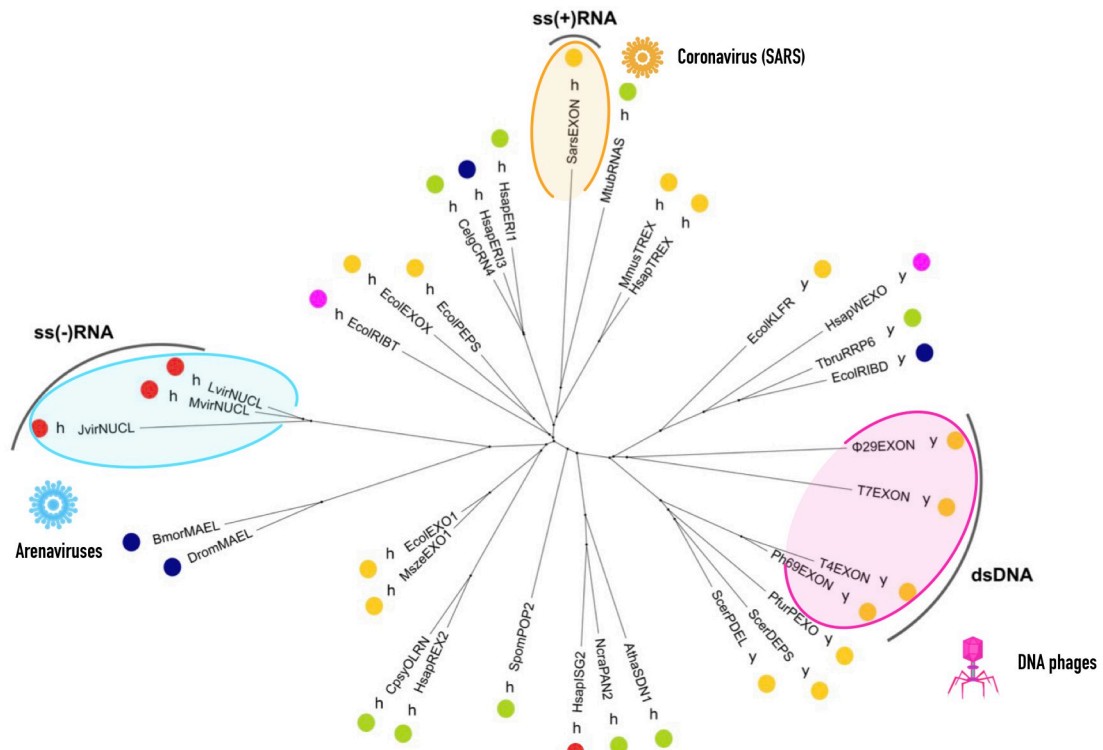

**Fig 1. Unrooted tree based on the structural comparisons of DEDD exonucleases.** The dots indicate exonuclease function, which colors stand for: yellow, proofreading; green, cytoplasmic RNA and/or DNA degradation; red, immune activity; dark blue, RNA processing and maturation; fuschia, DNA repair. The single letters indicate the DEDD exonuclease subgroup according to the fifth most conserved residue in the catalytic core: y, DEDDy subgroup; h, DEDDh subgroup. The tertiary structures of the exonucleases and their respective abbreviations used in this analysis were: DNA polymerase I klenow fragment, EcolKLFR; Exonuclease I, EcolEXO1; RB69 gp43 DNA polymerase, Ph69EXON; Phage T7 exonuclease, T7EXON; Phage T4 exonuclease, T4EXON; Phage phi29 exonuclease, φ29EXON; Nuclease domain of 3'hExo, HsapERI1; Human ISG20, HsapISG2; TREX2 3' exonuclease, HsapTREX; RNase D, EcolRIBD; WRN exonuclease, HsapWEXO; Pol III epsilon-hot proofreading complex, EcolPEPS; DNA-directed DNA polymerase, PfurPEXO; Pop2p deadenylation subunit, SpomPOP2; ERI1 exoribonuclease 3, HsapERI3; TREX1 exonuclease, MmusTREX; Cell-death related nuclease 4, CelgCRN4; DNA polymerase delta, ScerPDEL; Pan2 catalytic unit, NcraPAN2; Exonuclease X, EcolEXOX; nsp14 3–5 exoribonuclease, SarsEXON; Nucleoprotein with 3–5 exoribonuclease, LvirNUCL; RNase T, EcolRIBT; Mopeia virus Exonuclease domain, MvirNUCL; Junin virus nucleoprotein, JvirNUCL; Maelstrom of Drosophila melanogaster, DromMAEL; Maelstrom of Bombyx mori, BmorMAEL; Ribosomal RNA processing protein 6, TbruRRP6; RNase AS, MtubRNAS; DNA polymerase epsilon, ScerDEPS; Exonuclease I, MszeEXO1; Small RNA degrading nuclease 1, AthaSDN1; Oligoribonuclease, CpsyOLRN; and REXO2 oligoribonuclease, HsapREX2. For PDB IDs see supplementary data. Viruses figures (SARS and Arenavirus) were made using Keynote and the DNA phage was drawn by hand.

exonucleases have evolved diverse functions following three clearly independent viral hijacking events.

## Evolutionary analysis of the nidoviral ExoN domain

All the available data indicate that the nidoviral acquisition of the ExoN domain occurred prior to the diversification of these viruses and may have played a key role in their evolutionary success. As shown in Fig 2 and S1 Fig, all the nidoviruses with genomes larger than 20 kb are endowed with the ExoN domain, in which the catalytic pentad is highly conserved. As shown by the presence of the highly conserved cysteine and histidine residues, all the ExoNs of these RNA viruses have two ZFs. The only known exception is the ExoN domain of the Planarian secretory cell nidovirus (PSCNV) of the *Mononidoviridae* family, in which only one ZF can be

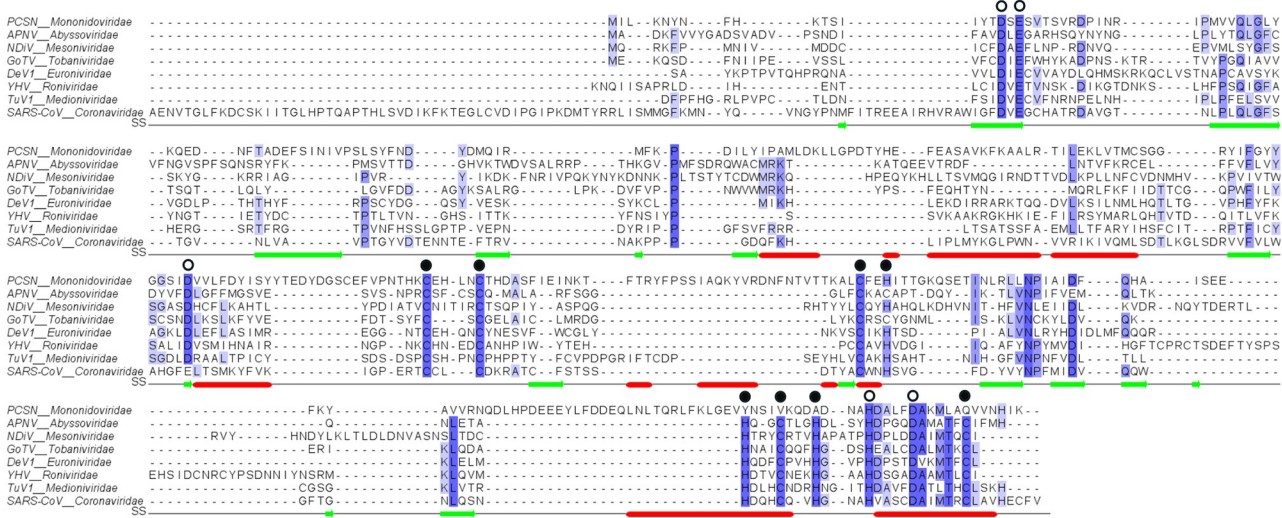

**Fig 2. Multiple sequence alignment of the nidovirus ExoN domain.** The Exo I (DE), Exo II (D/E), and Exo III (Dh) conserved sequence motifs are signaled with unfilled circles above them. Zinc binding motif 1 (ZF1, CCCH/C) and zinc-binding motif 2 (ZF2, HCHC) are signaled with filled circles above them. Secondary structure is indicated as red, helix; green, beta strand. Viral sequences abbreviation: PCSN, Planarian cell-secretory nidovirus; APNV, Aplysia californica nidovirus; NDiV, Nam Dinh virus; GoTV, Goat torovirus; DeV1, Decronivirus 1; YHV, Yellow head virus; and TuV1, Turrinivirus 1.

identified, while the residues that could correspond to an additional second ZF cannot be confidently assigned (Fig 2). In fact, the PSNCV has several major genomic and molecular differences with other nidoviruses, which might explain the absence of ZF2 (Table 2).

The *Nidovirales* ExoNs phylogenetic tree (Fig 3) shows each of the families forming its own clade, with high bootstrap values close to the edges. Only the families *Roniviridae* and *Euroniviridae* are clustered with high bootstrap values closer to the root of the tree. The Aplysia californica nidovirus (APNV) stems as a sister group to the *Coronaviridae* family, whereas the PSCNV is located at the root of the tree forming its own clade. The RdRp phylogenetic tree (Fig 3) exhibits a similar topology to the ExoNs tree, with high bootstrap values from the root to the edges. Most of the viral families form their own clade; however, some families are

**Table 2. Differences between PSCNV and other nidoviruses, based on Saberi et al., 2018 [14].**

|  | **Other Nidovirus** | **PSCNV** |
|---|---|---|
| **Genome size** | 12.7 to 35.9 kb | 41.1 kb |
| **Exonuclease** | Present, with the exception of *Amidovirineae* and *Nanidovirineae* Suborders. The ExoN domain contains two zinc fingers (ZF1 and ZF2). | Present, but instead of the characteristic 2H2C domain of nidovirus, the ExoN has a ES2Q domain and probably lacks ZF2. |
| **ORF1b size** | Nidovirus lacking ExoN 12.7–15.7 kb. | Disproportionately large including unannotated domains. |
|  | Nidovirus with ExoN 19.9–33.5 kb. |  |
| **ORF organization** | Multi-ORF arrangement. Overlapping ORF1a and ORF1b and multiple ORF's at the 3'-end (3´ORFS). | A single ORF overlapping other small ORF's in distinct reading frames. |
| **Additional genes** | Lack of ribonuclease T2, ankyrin and fibronectin type II genes. | Contains genes for ribonuclease T2 homolog, ankyrin and two fibronectins type II. |
| **3CLpro differences** | 3CLpro with cysteine as the catalytic nucleophile. | 3CLpro with Ser-His-Asp as a catalytic triad. |
|  | Substrate-binding pocket with a Histidine. | Substrate-binding pocket with a Valine. |
| **RdRp C motif** | Ser residue in the nidovirus-specific | Ser is replaced by a Gly residue in this signature (GDD). |
|  | SDD signature. |  |
| **NiRAN domain** | All nidoviruses retain seven invariant residues. | Only six of seven invariant residues are conserved. |

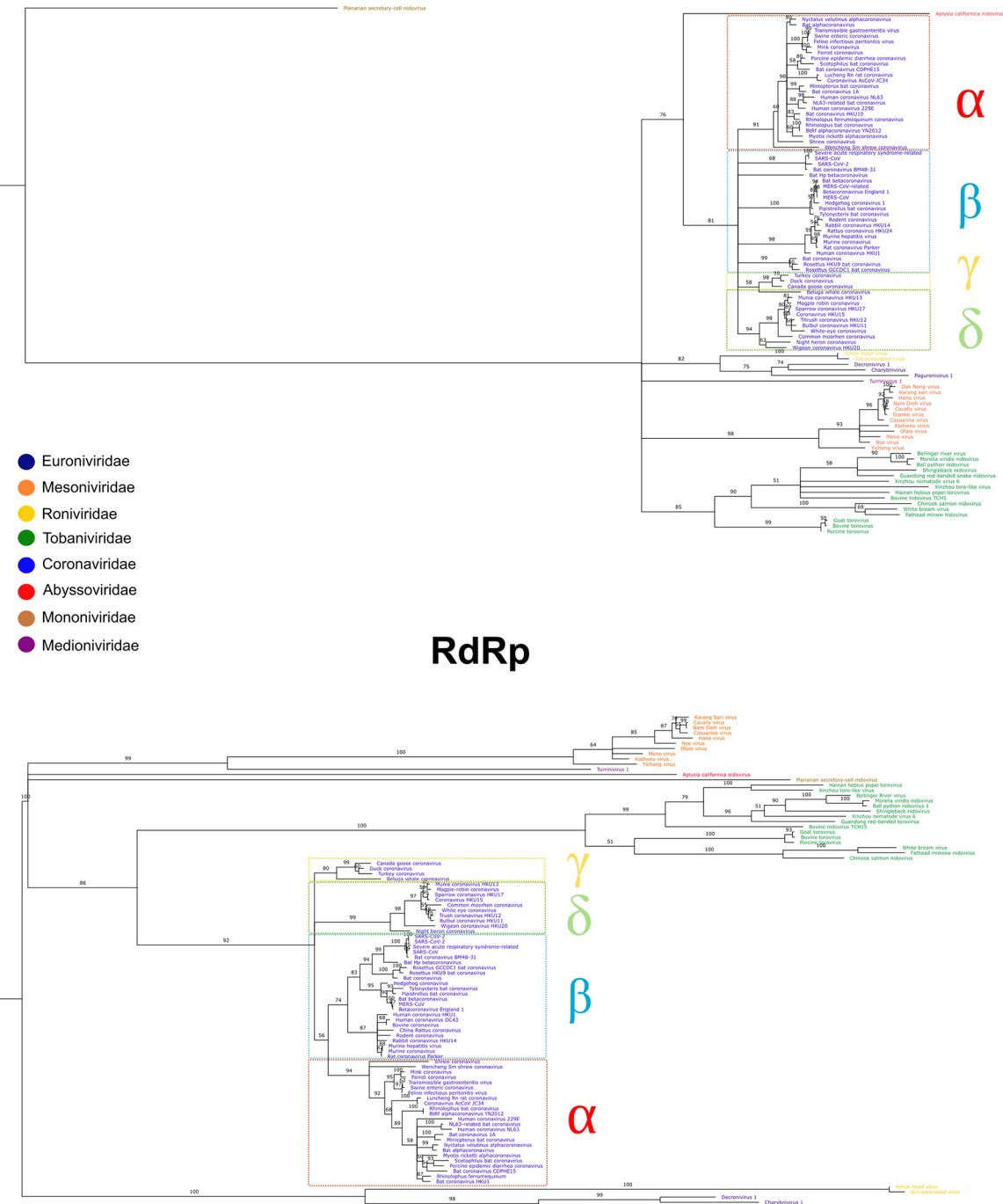

**Fig 3. Nidovirus RdRp and ExoN maximum-likelihood phylogenies.** The colors in the names of the viral species correspond to the families they belong to. The *Coronaviridae* family has been divided in its corresponding genuses: alphacoronavirus (α), betacoronavirus (β), deltacoronavirus (δ), and gammacoronavirus (ɣ). Bootstraps branches with < 50% have been collapsed.

grouped as sister groups, i. e. the *Coronaviridae* with the *Tobaniviridae*, the *Mesoniviridae* with the *Medioniviridae*, and the *Roniviridae* with the *Euroniviridae*. In this tree, the APNV as well as the PSCNV stem as independent clades. The overall topology is quite similar in both trees, with the different viral families consistently grouped.

The patterns of evolutionary relatedness among viral groups in both the RdRp and ExoN phylogenies exhibit a very similar topology. The high level of sequence conservation (95% identity) between SARS-CoV and SARS-CoV-2 ExoNs is similar to the one observed when the proteins involved in the RdRp complex (nsp8, nsp9, and nsp12) are compared (96% identity) [47]. The very high level of similarity between the topology of the ExoN tree with that of the highly conserved RdRp is an indication that even in RNA-based entities like nidoviruses, proofreading processes play a key role in maintaining genome integrity and stability.

## ZFs are present in RNA viral ExoNs, but not in cellular and phage enzymes

A detailed analysis of the DnaQ-like family of exonucleases structures showed that the only two cellular exonucleases with ZF domains are the "cell death-related nuclease 4" [48], and the "target of Egr1" [49]. The multiple sequence alignment of viral and cellular DnaQ-like exonucleases demonstrates the lack of ZFs in both the DNA phages and cellular ExoN domains (S2 Fig). In contrast with cellular DnaQ-like exonucleases, RNA viral exonucleases such as the SARS-CoV nsp14 ExoN and Lassa virus NP exonuclease are endowed with two and one ZFs, respectively (Fig 4A and 4B) [18, 50]. Interestingly, the peculiar Maelstrom (which has the typical ExoN β1-β2-β3-αA-β4-αB-β5-αC catalytic core topology but lacks the DEDD sequence motif) possesses a different and characteristic ZF (ECHC) in the exonuclease domain, which is also found in the arenaviral ExoN (Fig 4C) [27, 51]. These structural similarities and their positions in the structured-based tree suggest a rather close evolutionary relationship between arenaviral ExoN and Maelstrom (see below). No ZFs in the proofreading ExoN domains of Escherichia phage RB69 (*Myoviridae*), Bacillus virus phi29 (*Podoviridae*), Escherichia virus T7 (*Autographiviridae*), nor Escherichia virus T4 (*Myoviridae*) were identified (S2 Fig). Overall, the differences among viral ExoNs and the fact that they are located in different branches in our structure-based tree supports the idea of independent viral hijacking events.

## Discussion

### The viral ExoN domains

Apart from nidoviruses, the other only known RNA viruses endowed with a 3'-5' ExoN domain are the arenaviruses, which are enveloped, segmented negative-stranded RNA viruses that belong to the *Arenaviridae* family (Order *Bunyavirales*). However, it has been suggested that the arenaviral ExoN is involved in suppressing innate immune signaling and not in proofreading activity [52], which is consistent with the role of ExoN in the genome size increase of coronavirus and arenaviral small size genomes (10.5 kb). Although the SARS-CoV (*Coronaviridae*) and Lassa virus (*Arenaviridae*) ExoNs are homologous and possess features of the DnaQ-like family of exonucleases, structural differences between the proteins support the idea of independent acquisition events by these viruses, which in the case of the arenaviruses led to a different function. Several differences can be noted (Fig 4A and 4B). Firstly, the Lassa ExoN has a basic loop motif (K516, K517, K518, and R519) in a projecting "arm" located to the left of the active site that interacts with the non-substrate strand, and which is absent in SARS-CoV ExoN. Secondly, the SARS-CoV ExoN is endowed with an additional co-factor binding site (nsp10 binding-site) that is absent in the Lassa ExoN. Thirdly, the SARS-CoV ExoN is structurally interconnected to a N7-MTase domain, while the Lassa virus ExoN is linked to the NP core domain. Finally, the Lassa virus ExoN has one zinc finger with an as yet undescribed role,

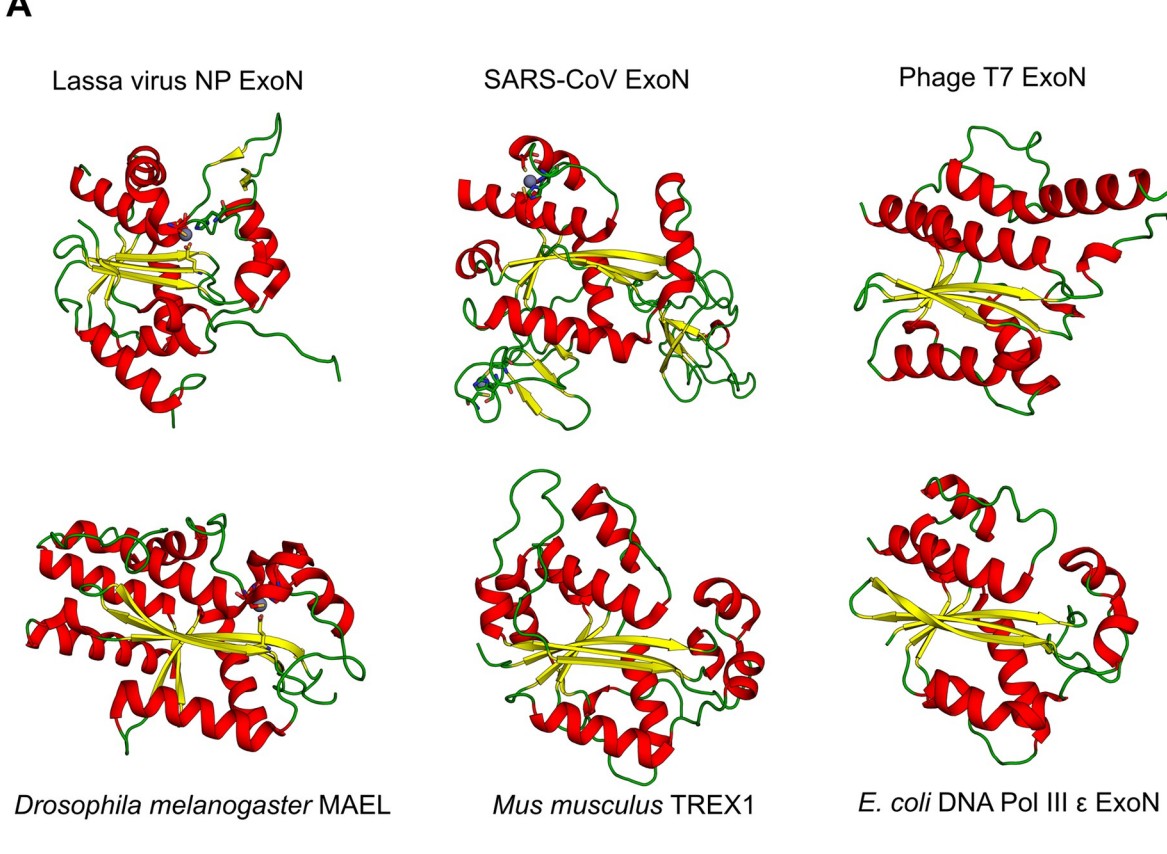

**A**

Lassa virus NP ExoN       SARS-CoV ExoN       Phage T7 ExoN

*Drosophila melanogaster* MAEL       *Mus musculus* TREX1       *E. coli* DNA Pol III ε ExoN

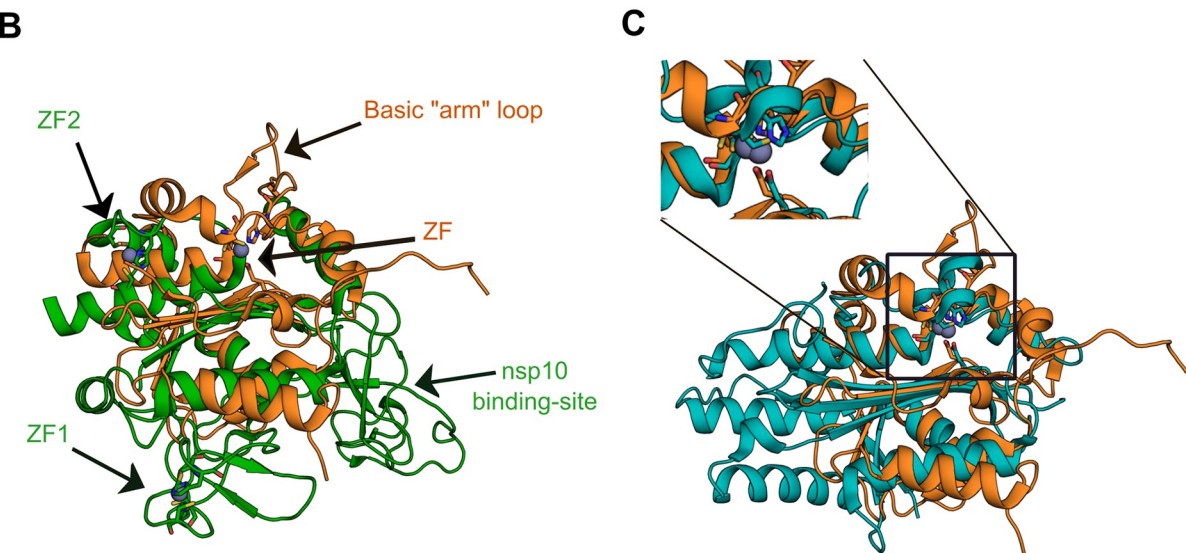

**B**

ZF2
Basic "arm" loop
ZF
ZF1
nsp10 binding-site

**C**

**Fig 4. Structural comparisons of DnaQ-like exonucleases.** (A) Structural conservation among the DnaQ-like nucleases. α-helices colored in red; β-sheets in yellow; and connecting elements in green. (B) Structural superposition between Lassa virus NP exonuclease (PDB ID: 3Q7V), orange; and SARS-CoV ExoN domain (PDB ID: 5C8U), green. (C) Structural superposition between Lassa virus NP exonuclease (PDB ID: 3Q7V), orange; and *Drosophila melanogaster* Maelstrom (PDB ID: 4YBG), blue. Herein, the presence of the ZF (ECHC) in both proteins is highlighted.

whereas the SARS-CoV ExoN is endowed with two zinc fingers involved in structural stability (ZF1) and possibly in enzymatic activity (ZF2) [18, 27, 50].

As shown in Fig 1, several dsDNA phages have a proofreading ExoN domain that shares with DnaQ-like exonucleases the well-conserved DEDD motif, the 3'-5' exonucleolytic degradation of DNA, and the β1-β2-β3-αA-β4-αB-β5-αC conserved catalytic core topology [23–26, 28]. In these DNA phages, the ExoN domain is covalently linked to their polymerase, as in *Escherichia coli* DNA polymerase I and III, suggesting that both the polymerase and the ExoN were taken in a single hijacking event by the ancestor of the DNA phages. This indicates an independent hijacking event different from the ones of the arenaviral and nidoviral ExoNs, which is consistent with their different locations in the structure-based tree shown in Fig 1, are consistent with the idea of a distinct viral hijacking event for the phages ExoNs. This is further supported by the fact that they belong to the DEDDy subgroup rather than the DEDDh subgroup [53], where SARS-CoV and arenaviral ExoNs are located.

## The arise of zinc fingers at the ExoN domain in nidoviruses

The groundbreaking discovery of catalytic RNAs in the early 1980s gave considerable credibility to the proposal that the first living entities were based on RNA as both the genetic material and as catalyst, a hypothetical stage called the RNA world [54]. Indeed, the catalytic, regulatory, and structural properties of RNA molecules and ribonucleotides, combined with their ubiquity in cellular processes, suggest that they played a key role in early evolution and perhaps in the origin of life itself [55, 56]. Today the only known RNA-based biological entities are found in the wide array of RNA viruses and viroids. Although RNA viruses may provide insights into the structure and evolution of early cellular genomes prior to the emergence of DNA, it is unlikely that they are direct descendants of primitive RNA-based life forms [57]. The viral ability to cross taxonomic barriers and infect new species is well established, but all the available evidence indicates that nidoviruses are restricted to the *Animalia*, suggesting that the *Nidovirales* order originated late in the history of the biosphere. Since the appearance of animals occurred sometime around 750 million years ago, the available data indicate that the nidoviral hijacking of the ExoN domain took place during the late Proterozoic.

Our results confirm and extend the conclusions of Snijder et al. (2003) that cellular and DNA phages of the DnaQ-like exonucleases lack ZFs within the exonuclease domain (S2 Fig). This stands in sharp contrast with the nidoviral enzyme, in which ZFs appear to play an essential role in structure stability and perhaps also in catalysis [18]. Experimental analyses reinforce this conclusion. Mutagenesis studies targeting ExoN zinc finger 1 (ZF1) from Murine hepatitis virus (C206A and C209A), Transmissible gastroenteritis virus (C210H) and MERS-CoV (C210H) are known to affect genome replication [20, 21, 58]. Additionally, White bream virus (C6101A, C6104A, C6122A, and C6125A), and SARS-CoV ExoN ZF1 mutants were found to lack nucleolytic activity and cannot be expressed as soluble proteins, respectively [18, 59]. Furthermore, ExoN ZF2 mutants for SARS-CoV (C261A and H264R), and MERS-CoV (C261A and H264R) abolished enzymatic activity and abrogated replication, respectively [18, 21]. The conservation of ZFs across nidoviruses (Fig 2) and the mutagenesis studies mentioned above indicate that they play a key structural role in viral ExoN function. Sequence conservation and ZF traits suggest a monophyletic acquisition of the ExoN domain that took place in the ancestral nidovirus population prior to its split into several families.

As mentioned above, with the exception of the PSCNV, all known nidoviruses with genomes larger than 20 kb possess two ZFs. The PSCNV is an interesting case and warrants further experimental analysis. On the one hand, it is the RNA virus with the largest linear genome characterized as of today, with a length of 41.1 kilobases [14], which is remarkable for

RNA-based biological entities with such an elevated mutation rate [60]. Nevertheless, PSCNV ExoN seems to lack one of the two ZFs, more specifically, ZF2, which has been shown to be essential for the correct function of the ExoN and replication, at least for two betacorona-viruses [18, 21]. These observations question the nature of this virus, which may not be a typical nidovirus, and the mechanism by which this unusually large RNA virus can keep its genome integrity and stability.

## The possible cellular origin of MAEL domain

Maelstrom (MAEL) is a conserved endoribonuclease present in metazoans and protists. It is related to the regulation of certain endogenous genetic elements such as retrotransposons [51, 61, 62]. Mutation assays that reduce the activity among MAEL orthologs indicate that MAEL is involved in ssRNA binding and not in catalysis. In particular, MAEL has been described as an RNA-binding protein that interacts with piwi-RNAs (pRNA), protecting the genome from transposons by repressing them in animal gonads [51, 63]. MAEL seems to be highly similar in structure to the arenaviral NP ExoN, but lacks the DEDD sequence motif (Figs 1 and 4C). However, evolutionary studies by Zhang et al. [61] demonstrated that the MAEL domain in protists such as *Entamoeba histolytica*, *Trypanosoma brucei*, and *Leishmania braziliensis* is endowed with the DEDD motif. Chen et al. [64] have reported the existence of a MAEL domain in the amoeba that has both sequence motifs (**DEDD** and an **ECHC** MAEL-tetrad) and a potential exoribonuclease activity. The structural similarity between the arenaviral exo-nuclease and MAEL, supports the possibility of a cellular origin of the arenaviral exonuclease domain. As suggested by Sato and Siomi [63], MAEL may have evolved from a DEDDh exonu-clease to an ECHC tetrad only by switching the catalytic residues. Thus, the arenaviruses could have hijacked the enzyme prior to the emergence of the RNA-binding moonlighting function and the loss of enzymatic activity as an exoribonuclease. A possible mechanism for protein evolution in nucleases suggested by Ballou et al. [65] following Jeffery's [66] proposal could explain this evolutionary transition reinforcing the hypothesis of its ultimate cellular origin.

## Conclusions

The results presented here suggest that ExoN genes have been hijacked by viruses at least three times: once by DNA phages (RB69, φ29, T4, T7) and, independently, by arenaviruses and by nidoviruses, both of which are RNA viruses. The presence of ExoN led to a major increase in nidovirales genomes, which are endowed with the biggest viral RNA genomes known. The PSCNV is an odd case and further studies may shed light on its evolutionary history. The case of the cellular MAEL enzyme is, on the other hand, quite impressive; the exonuclease-like domain folding changed its function across the evolution from DNA edition to RNA-binding, losing its catalytic activity in the process.

As shown in Fig 1, ExoN activity is quite unspecific and includes both RNA and DNA sub-strates. It is reasonable to assume that the early evolution of exonucleases represented a critical step in enhancing the encoding capabilities of primitive RNA genomes. It may not be so diffi-cult to evolve exonuclease activity–after all, it involves a simple hydrolase reaction capable of destroying a phosphodiester bond in a genetic polymer already strained due to a mismatched base-pair. Thus, the postulated transition from small, fragmented RNA to much larger DNA cellular genomes would have been facilitated by the lack of absolute substrate specificities of ExoN [67].

The RNA proofreading activity by the ExoN domain in nidoviruses and the immune eva-sion function of the arenaviral ExoN highlights the versatility of these enzymes, in which a few structural changes can lead to a novel function. The conservation of the residues that form the

ZFs and coordinate the metal ions show the importance of these motifs in the structural stabilization of exonucleases in RNA-based entities such as arenaviruses and nidoviruses. Finally, the presence and the importance of ZFs in the RNA viral exonucleases analyzed in our work opens the possibility of developing antiviral therapies using zinc chelating agents.

## Supporting information

**S1 Fig. Sequence conservation among nidoviral ExoNs.** In SARS-CoV ExoN structure, Exo I (DE), Exo II (D/E), and Exo III (D) conserved sequence motifs are highlighted in green. Zinc-binding motif 1 (ZF1, CCCH/C) and zinc-binding motif 2 (ZF2, HCHC) are highlighted in red. $Zn^{2+}$ is depicted as dark grey spheres and $Mg^{2+}$ as a yellow sphere. In the logo, Exo I, Exo II, and Exo III are signaled with green arrows, while ZF1 and ZF2 are signaled with red arrows. Logo was made with WebLogo 3 (http://weblogo.threeplusone.com/).
(TIF)

**S2 Fig. Cellular and viral exonucleases alignment.** The Exo I (DE), Exo II (D/E), and Exo III (D) conserved sequence motifs are signaled with unfilled circles. Zinc-binding motif 1 (ZF1, CCCH/C) and zinc-binding motif 2 (ZF2, HCHC) are signaled with filled circles. Due to experimental procedures, the DEDD catalytic residues of the Bacillus virus phi29 ExoN domain (PhagePhi29) were mutated, showing a AADD sequence motif (the mutated residues are depicted with gaps in this alignment).
(TIF)

**S1 Table. PDBs of the DnaQ-like exonucleases used in this work.**
(DOCX)

## Acknowledgments

WC-S is a doctoral student from the Posgrado en Ciencias Biológicas, Universidad Nacional Autónoma de México (UNAM) and received fellowship CVU-815057 from CONACyT. This article is part of the doctoral thesis of IM-V which received fellowship 415961 from CONACyT. AC-G received fellowship from CONACyT (CVU-1002377).

## Author Contributions

**Conceptualization:** Adrián Cruz-González, Israel Muñoz-Velasco, Wolfgang Cottom-Salas, Arturo Becerra, José A. Campillo-Balderas, Ricardo Hernández-Morales, Alberto Vázquez-Salazar, Rodrigo Jácome, Antonio Lazcano.

**Investigation:** Adrián Cruz-González, Israel Muñoz-Velasco, Wolfgang Cottom-Salas, Arturo Becerra, José A. Campillo-Balderas, Alberto Vázquez-Salazar, Rodrigo Jácome, Antonio Lazcano.

**Methodology:** Ricardo Hernández-Morales.

**Writing – review & editing:** Adrián Cruz-González, Israel Muñoz-Velasco, Wolfgang Cottom-Salas, Arturo Becerra, José A. Campillo-Balderas, Ricardo Hernández-Morales, Alberto Vázquez-Salazar, Rodrigo Jácome, Antonio Lazcano.

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
