## [Decision Letter · Decision Letter 0]

25 Feb 2021

Structural analysis of viral ExoN domains reveals polyphyletic hijacking events

PONE-D-21-03009

Dear Dr. Lazcano,

We’re pleased to inform you that your manuscript has been judged scientifically suitable for publication and will be formally accepted for publication once it meets all outstanding technical requirements.

Kind regards,

Jean-Luc EPH Darlix, MG, Ph.D.

Academic Editor

PLOS ONE

Reviewers' comments:

Reviewer's Responses to Questions

**Comments to the Author**

1. Is the manuscript technically sound, and do the data support the conclusions?

Reviewer #1: Yes

2. Has the statistical analysis been performed appropriately and rigorously? 

Reviewer #1: Yes

3. Have the authors made all data underlying the findings in their manuscript fully available?

Reviewer #1: Yes

4. Is the manuscript presented in an intelligible fashion and written in standard English?

Reviewer #1: Yes

5. Review Comments to the Author

Reviewer #1: This is a nice, concise study of structural similarities between viral exonucleases. It has clear implications for SARS-CoV-2 researchers. It should be of broad interest to readers. I recommend publication.

6. PLOS authors have the option to publish the peer review history of their article (what does this mean?). If published, this will include your full peer review and any attached files.

Reviewer #1: No

---

## [Editor Report · Acceptance letter]

2 Mar 2021

PONE-D-21-03009 

Structural analysis of viral ExoN domains reveals polyphyletic hijacking events 

Dear Dr. Lazcano:

I'm pleased to inform you that your manuscript has been deemed suitable for publication in PLOS ONE. Congratulations! Your manuscript is now with our production department. 

Kind regards, 

on behalf of

Professor Jean-Luc EPH Darlix 

Academic Editor

PLOS ONE